# Treatment of Complex Regional Pain Syndrome in Children and Adolescents: A Structured Literature Scoping Review

**DOI:** 10.3390/children7110245

**Published:** 2020-11-20

**Authors:** Andrea Vescio, Gianluca Testa, Annalisa Culmone, Marco Sapienza, Fabiana Valenti, Fabrizio Di Maria, Vito Pavone

**Affiliations:** Department of General Surgery and Medical Surgical Specialties, Section of Orthopaedics and Traumatology, Surgery, AOU Policlinico-Vittorio Emanuele, University of Catania, 95123 Catania, Italy; andreavescio88@gmail.com (A.V.); gianpavel@hotmail.com (G.T.); annalisa.culmone@libero.it (A.C.); marcosapienza09@yahoo.it (M.S.); valentifabiana@gmail.com (F.V.); fdimaria95@gmail.com (F.D.M.)

**Keywords:** pediatric, growing age, complex regional pain syndrome, reflex sympathetic dystrophy, multidisciplinary, physical therapy, cognitive behavioral therapy, drugs, pharmacological treatment, occupational therapy

## Abstract

Background: Complex regional pain syndrome (CRPS) is characterized by chronic, spontaneous and provoked pain of the distal extremities whose severity is disproportionate to the triggering event. Diagnosis and treatment are still debated and multidisciplinary. The purpose of this systematic review is to analyze the available literature to provide an update on the latest evidence related to the treatment of CRPS in growing age. Methods: Data extraction was performed independently by three reviewers based on predefined criteria and the methodologic quality of included studies was quantified by the Newcastle–Ottawa Quality Assessment Scale Cohort Studies. Results: At the end of the first screening, following the previously described selection criteria, we selected n = 103 articles eligible for full-text reading. Ultimately, after full-text reading and a reference list check, we selected n = 6. The articles focused on physical (PT), associated with cognitive behavioral (CBT) and pharmacological (PhT) treatments. The combination of PT + CBT shows the most efficacy as suggested, but a commonly accepted protocol has not been developed. Conclusions: Physical therapy in association with occupational and cognitive behavioral treatment is the recommended option in the management of pediatric CPRS. Pharmacological therapy should be reserved for refractory and selected patients. The design and development of a standard protocol are strongly suggested.

## 1. Introduction

First described in the 17th century as “causalgia” [1], complex regional pain syndrome (CRPS) is characterized by chronic, spontaneous and provoked pain of the distal extremities whose severity is disproportionate to the triggering event [2]. Three different CRPS subtypes have been distinguished:Type 1, previously known as reflex sympathetic dystrophy (RSD), whose cause is not always known.Type 2, which results from nerve damage.Type 3, or not otherwise specified CRPS, which partly shares clinical and diagnostic aspects with the previous types [2].

CRPS type 1 affects children and adolescents aged 5 to 17 years, with a peak incidence around the 13th year of age, and it is more frequently found in women (70% of cases) [3,4]. The pathogenic mechanism is still unclear, although several hypotheses have been proposed. Genetic factors, altered microcirculation and traumas, such as sprains, fractures and surgical procedures, contribute to pain symptoms. Anxiety, somatization and familial and school problems could also play a role. Chronic pain, generally unilateral and limb localized, autonomic and motor dysfunction and trophic disorders are the main symptoms of CRPS type 1. There are two presentations of the syndrome: the “warm” one, with red, warm, swollen skin that usually occurs in the acute phase, and the ‘‘cold’’ one, with blue/purple, cold, sweaty skin, which is usually associated with the chronic phase [3,4,5,6]. The diagnosis of CRPS type 1 is clinical and based on the Budapest diagnostic criteria [4]. However, diagnosis remains challenging due to the lack of validated diagnostic tests and the difficulty of differential diagnosis. Laboratory and imaging tests can be helpful in the event of diagnostic doubt [1]. Treatment is multidisciplinary, and it is mostly based on physical and psychological therapy and medications; only in selected subjects is treatment invasive [7]. The purpose of this systematic review was to analyze the available literature to provide an update on the latest evidence related to the treatment of CRPS type 1 in children and adolescents, highlighting the multidisciplinary approach.

## 2. Materials and Methods

### 2.1. Literature Search Strategy

A systematic review of the current literature was conducted according to the Preferred Reporting Items for Systematic Reviews and Meta-Analyses (PRISMA) guidelines [8]. On 20th September 2019, three independent authors (SM, VF and DiMF) performed a systematic review of two different medical electronic databases (PubMed and Web of Science). To achieve the maximum sensitivity of the search strategy, a search string was used (“(complex regional pain syndrome OR reflex sympathetic dystrophy OR Sudeck’s atrophy) AND (pediatric OR adolescent OR children OR childhood) AND (treatment OR management)”).

### 2.2. Selection Criteria

The reference lists of all retrieved articles were reviewed for further identification of potentially relevant studies, and the articles were assessed using the inclusion and exclusion criteria. The following inclusion criteria were used when screening titles and abstracts: (a) studies of any level of evidence; (b) studies written in the English language; (c) studies reporting clinical or preclinical results; (d) published studies in peer review journals on the treatment of complex regional pain syndrome type 1. The exclusion criteria were as follows: (a) review articles, (b) case reports, (c) articles written in other languages, (d) diagnosis or differential diagnosis of complex regional pain syndrome type 1. We also excluded all the remaining duplicates, articles dealing with other topics and those with poor scientific methodology or without an accessible abstract. Reference lists were also hand-searched for further relevant studies.

### 2.3. Data Extraction and Criteria Appraisal

All data were extracted from article texts, tables and figures. Three investigators (SM, VF and DMF) independently reviewed each article. Discrepancies between the three reviewers were resolved by discussion and consensus. The final results and any remaining controversy on the reviewed article were reviewed and discussed with the senior investigators (VA and CA), who served as independent reviewers and assessed study quality. Conflicts about data were resolved by the senior surgeon (PV). Reference lists from the selected papers were also screened. The PRISMA flowchart for the selection and screening method is provided in Figure 1.

### 2.4. Risk of Bias Assessment

A risk of bias assessment of all selected full-text articles was performed according to the Newcastle–Ottawa Quality Assessment Scale Cohort Studies (NOS) [9]. The NOS contains eight items, categorized into three dimensions including selection, comparability and—depending on the study type—outcome (cohort studies) or exposure (case-control studies). For each item, a series of response options is provided. A star system is used to perform a semi-quantitative assessment of study quality, such that the highest quality studies are awarded a maximum of one star for each item, except for the item related to comparability, which allows the assignment of two stars. The NOS ranges between zero and nine stars. The assessments were performed by two authors (VA and CA) independently. Any discrepancy was discussed with the senior investigator for the final decision. All the raters agreed on the final result of every stage of the assessment (Appendix A). In the systematic review, studies classified with more than six stars were included.

## 3. Results

### 3.1. Study Selection

From the search of PubMed and Web of Science, 264 articles were included in the review, and 24 studies were selected after duplicate exclusion. Following the inclusion and exclusion criteria, the first screening was performed. A total of 103 papers were considered eligible. Finally, after the full-text reading, reference list check and risk of bias assessment, six studies were included. A PRISMA [8] flowchart of the method of selection and screening is provided (Figure 1).

The main focus of the included studies was related to physical (PT), cognitive behavioral (CBT) and pharmacological (PhT) treatments. A summary of the results is provided in Table 1.

### 3.2. Physical Therapy and Cognitive Behavioral Treatment

Three included studies contained a combination of physical therapy and cognitive treatment. Sherry et al. [10] reported the complete resolution of pain symptoms in 74.7% of the sample. Seven subjects did not have remission. One child was dysfunctional with CRPS pain, and five had persistent mild pain but were fully functional. The authors highlighted suicide attempts (*p* = 0.026), an eating disorder (*p* = 0.028), reporting less pain initially (*p* = 0.021) and scoring higher on the Brief Symptom Inventory subsets for depression and paranoid ideation (*p* 0.037 and 0.048, respectively) as predictors of recurrence. Lee et al. [11] divided patients into two different groups: PT + CBT for 3 weeks vs. PT + CBT for 6 weeks. Both the cohorts showed improvement in all pain and physical functioning outcome measures with short- and long-term follow-up, without differences in pain scores, recurrent episodes of CRPS or participation in school or activities. Logan et al. [12] assessed 56 patients aged 8–18 years at admission and at discharge, and every parameter evaluated had statistically significant improvements. Thirty-two percent of patients required an assistive device at admission, while none required one at discharge.

### 3.3. Pharmacological Treatment

Petje et al. [13] assessed functional outcome in patients treated with an intravenous infusion of iloprost, a prostacyclin analogue. The drug was administered 6 h per day on 3 consecutive days. Among the side effects noted were headache on the first day of infusion in all patients and flushing and vomiting on the second day in three patients. A decrease in systolic blood pressure of an average of 7 mm Hg (5–15 mm Hg) in the first 30 min after administration was detected in almost the entire cohort. Improvement of the visual analogue scale (VAS) score was found (*p* < 0.05). Relapse of CPRS was experienced by two patients, the first after 3 months. Brown et al. [14] analyzed the outcome of 29 patients refractory to PT + CBT. Fourteen subjects underwent amitriptyline administration, and 15 underwent gabapentin administration. After the 6-week trial, both cohorts showed improvement in pain symptoms, sleep disturbances and functionality. No statistically significant differences were found (*p* = 0.77) between the drugs. Similar adverse events were recorded (*p* = 0.77). Donado and colleagues [15] investigated the use of continuous regional anesthesia (epidural or peripheral catheter) in subjects refractory to PT + CBT. Their data showed significant changes between admission and discharge for pain (*p* < 0.0001), without significant changes throughout the 4-month period after admission (*p* > 0.05).

## 4. Discussion

According to our data, the physical therapy combined with cognitive behavioral treatment should be considered as the most appropriate first-line approach in CRPS-affected children and adolescents. The pharmacological therapy was found to be efficient in the PT + CBT failure case; the use of drugs is useful only in selected patients subjected to adjuvant physical and cognitive protocol. Nowadays, the lack of comparison treatment studies and specified outcome measurements does not allow for more detailed analysis or the development of a standard treatment.

CRPS is a common but not completely understood disorder, with no available data about the incidence of pediatric CRPS [5] because the diagnosis is uncertain and underestimated. Early diagnosis is as important as treatment; in fact, a longer disease course and sequelae [2] are associated with late identification. Unfortunately, no specific diagnostic tools have been developed for children and adolescents, so the adult criteria are used [5]. Orthopedists have a key role in the recognition of the disease due to very little evidence, no common consensus among the physicians and a lack of guidelines [6]. As reported by Berde and Lebel [16], often, the choice of treatment may vary according to the experience and resources of the clinician. Several treatments have been described [2], including acupuncture, transcranial magnetic stimulation and invasive procedures, but the efficacy has been proven for the combination of physical and cognitive behavioral therapy and only pharmacological treatments. The most established treatment is a program of physical rehabilitation and cognitive behavioral therapy [15]. The goal is to restore normal function, increase the joint motion range and load tolerance and strength and concurrently assist the child in accepting and managing the pain [2].

Different protocols were illustrated in selected studies, highlighting the absence of a standard treatment protocol. Sherry and colleagues [10] suggested aerobic training and progressive resistive exercises, in addition to hydrotherapy, desensitization with towel rubbing, hand massage, textured fabric rubs and contrast baths (2 and 38 °C). During the patient’s hospitalization, 5–6 h of daily exercise therapy and 45 min to 3 h of home exercise program (HEPs) were performed. In the Logan et al. trial [12], the patients underwent open-chain and closed-chain activities and an individualized HEP, and each child’s functional goals, such as playing a specific sport, were incorporated into the physical schedule. In addition, this multidisciplinary rehabilitation approach addresses the entire pain experience, incorporating desensitization, exposure to feared activities, skills for coping with pain and changes to social responses to pain. Lee et al. [11] designed a protocol including transcutaneous electrical nerve stimulation, progressive weight bearing, tactile desensitization, massage, contrast baths and an HEP. Six weekly sessions of individual CBT incorporating pain management strategies, including relaxation training, deep breathing exercises, biofeedback and guided imagery, were also included. Patient compliance, nurse care and parent treatment programs [17] are crucial to promote successful remission from pain and restoration of functional ability. Despite a rigorous rehabilitation program, Sherry et al. described their patients as a motivated and eager to please sample [10]. Lee et al. [11] recorded compliance varying between 78% and 82%. No adverse events were recorded in the three studies, but the remission rate varied between 79% and 100% [10,11,12]. Several authors have investigated the role of PT + CBT in pediatric CRPS type 1, especially the brain and neurological changes and treatment action on the central nervous system. Frot et al. found evidence of emotional integration of pain in CRPS patients [18]. Lebel et al. [19] concluded that some changes in the brain persist, especially in the amygdala and basal ganglia, even after symptomatic recovery [20]. Diers et al. [21] demonstrated that behavioral extinction training reduces the emotional involvement in processing painful stimuli and induces a shift to a more sensory-discriminative way of pain processing post-treatment. Kregel et al. [22] emphasize that conservative treatments for patients with chronic musculoskeletal pain may induce both functional and morphological changes in predominantly prefrontal brain regions. For these reasons, non-invasive treatments are often recommended, even in recurrent forms [2,4].

On the other hand, the literature presents evidence of good outcomes after intravenous infusion of drugs and regional nerve blockades. Three pharmacological trials were selected in the study, and different molecules were investigated. Petje et al. [13] assessed the outcome of iloprost intravenous infusion, an analogue of prostacyclin, which induces transitory complete sympathicolysis and avoids the anxiety associated with a lumbar sympathetic blockade. Despite the good rate of response, relevant adverse reactions such as headache, flushing, vomiting and a decrease in systolic blood pressure were recorded in all cohorts; consequently, the same authors do not suggest iloprost as primary therapy. Other drugs proposed for treating neuropathic pain were gabapentin and amitriptyline. The first avoids the release of neurotransmitters acting on voltage-gated calcium channels [23], and the latter neuropathically reinforces the serotonin transporter [24]. Brown et al. compared the two molecules in a refractory PT + CBT schedule. The series revealed that both drugs are effective in reducing pain scores and improving sleep, without significant differences. However, ventricular conduction abnormalities were noted in the gabapentin group, while amitriptyline was linked to QT prolongation, torsade de pointes and sudden cardiac death. For these reasons, the use of both drugs in selected patients and with proper monitoring may be considered. Several invasive options have been proposed, including the use of continuous regional anesthesia with epidural or peripheral catheters, which demonstrate a reduction in pain score and improvement in function score at short- and long-term follow-up. On the other hand, in the Donado et al. series [15], 39% of the sample did not experience clinical improvements in pain symptoms, and 43% had no functional advantages.

Nevertheless, the authors suggested the treatment in addition to an active PT + CBT protocol. All the studies included in the systematic review emphasized the utility of PT + CBT, even when additional approaches were undertaken. The comparison of management with versus without rehabilitation was considered ethically unacceptable [2]; however, the outcome may not be related to a single treatment, and the results have been influenced by conservative treatment even in pharmacologic protocol studies. Future research directions should focus on the identification of disease onset mechanisms and the development of more defined, proper and easy-to-use diagnostic tools.

The design of high-quality, prospective, large-cohort, long-term follow-up studies is strongly encouraged, as is the design of a specific assessment score.

Limits of the study are several and included the heterogeneity of the scores utilized in the objective clinical assessment of the patients. VAS score is an unspecific tool which aims to evaluate the pain, a limited feature of CPRS. At the same time, some authors used more specific scores, such as FDI and PFD, which were not initially developed for CRPS. In addition, due to the challenging diagnosis and long, individual and expensive treatments, studies by Brown et al. [14], Petje et al. [13] and Lee et al. [11] included small size samples, but the described protocols were evaluated with great interest for future prospective use. Moreover, the literature contains limited high-quality studies: despite major randomized prospective studies being included in the review, no control group article or direct comparisons between treatments have been published. The results of drug-related trials could be influenced by different factors, such as longer follow-up after the first or second PT + CBT treatment. The absence of standard protocols and the lack of randomized, blinded prospective trials are the main limits in the comparison of study results.

## 5. Conclusions

Complex regional pain syndrome in children and adolescents remains a challenge for the physician. The definition is not clear or commonly accepted, and this results in several undiagnosed cases. Despite several diagnostic standards being proposed in adults, no specific diagnostic criteria in growing age patients have been developed and proper treatment is often dependent on the physician’s experience and the treatment opportunities. A multidisciplinary approach is mandatory for a good outcome. Main findings of this review are represented by the consideration of physical, occupational and cognitive behavioral therapy as the first-line recommended options in the management of pediatric CPRS; pharmacological therapy can be utilized in failure cases. Unfortunately, the lack of a standard, less stressful and expensive protocol remains the main limit of the methodology. PhT often demands patient hospitalization and is reserved for selected subjects; the adverse events are common but considered as minor complications. Moreover, drugs or other treatments are not considered an alternative to the PT + CBT. To define proper diagnostic criteria, expert multidisciplinary committees as well as standard and commonly accepted guidelines and treatment protocol are essential and strongly encouraged. At the same time, the development of pilot studies for multicenter prospective trials could play a key role in the identification of more satisfactory treatments.

## Figures and Tables

**Figure 1 children-07-00245-f001:**
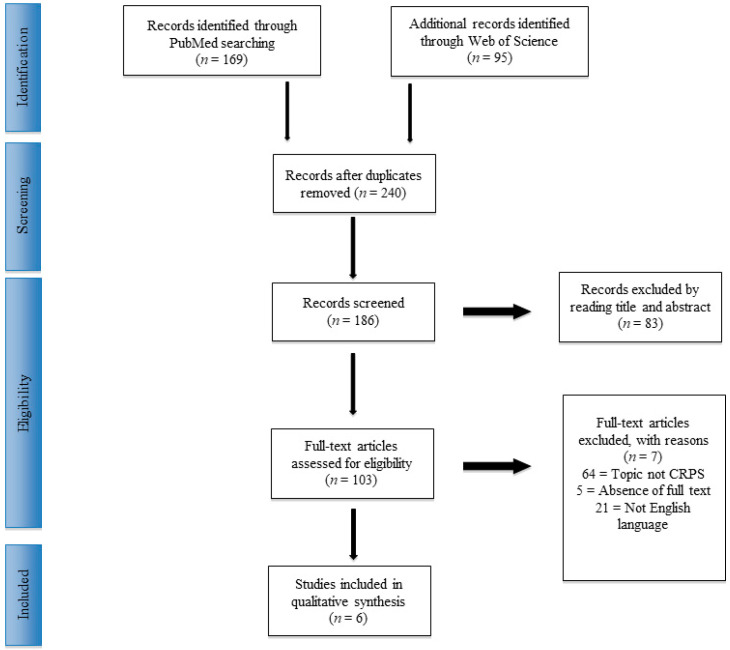
PRISMA (Preferred Reporting Items for Systematic Reviews and Meta-Analysis) flowchart of the systematic literature review.

**Table 1 children-07-00245-t001:** Results of selected studies.

Author	Subjects	Dignosis Criteria	Assessment	Treatment	Results	Limits
Brown et al. 2016	Amitriptyine Group: n = 14; Garbapentin Group: n = 15.	Modified International Association for the Study of Pain (IASP) clinical and research criteria.	Coloured Analogue Scale (CAS) Pain 6-weeks post-trial start; Sleep disability as measured on an internally developed 5-point Likert scale; Adverse events.	Amitriptylin 10 mg (at bedtime). Gabapentin at 900 mg/d (300 mg three times per day.	CAS *p* = 0.77. Sleep *p* = 0.26. Adverse events *p* = 0.75.	Small sample size. No randomization. No placebo group. No medium- and long-term follow-up.
Petje et al. 2003	n = 7	Skin examination; burning, dysesthesia, paresthesia and hypalgesia to cold. Skin cyanosis, mottling, hyperhidrosis, edema and coldness of the extremity and muscles, joint affliction duo to muscle hypotrophy or atrophy and range of motion (ROM) of the joints in the involved extremity. Bonica classification.	Visual analog scale VAS (0–10 points).	Intermittent intravenous infusion of Iloprost at 2 ng/kg/minute for approximately 6 h per day on 3 consecutive days + physiotherapy and psychologic.	VAS = *p* < 0.05. All patients had headache at the first day of infusion. 3 patients had flushing. 2 patients had vomiting. 86% of the sample had a decrease in systolic blood pressure with an average of 7 mm Hg (5–15 mm Hg) in the first 30 min after administration of Iloprost.	No control group Retrospective series. Small number of cases.
Donado et al. 2017	n = 102	Modified International Association for the Study of Pain (IASP) clinical and research criteria.	Preadmission, discharge and 4-month follow-up Pain Score (PS), Pain-Related Functional Disability (PFD) and sleep disturbances (SD)	Continuous regional anesthesia (epidural or peripheral catheter).	PS preadmission median = 7.0; IQR, 5.8–8.2. PS discharge = 3.1; IQR, 1.5–5.4; (*p* < 0.0001). PS 4-month follow-up = 4.3; IQR, 2.0–6.0; (*p* < 0.0001). PFD at admission had a moderate positive correlation with PDF at discharge (r, 0.5; *p* < 0.0001) SD = Yes 48.04%.	Retrospective design. Completed a full course of cognitive behavioral therapy
Logan et al., 2012	n = 56	Modified IASP clinical and research criteria.	At admission and at discharge. Numeric rating Scale (NRS). Functional Disability Inventory (FDI). Lower extremity functional scale: (LEFS). Canadian Occupational Performance Measure: (COPM). Multidimensional Anxiety Scale for Children (MASC). Children’s Depression Inventory (CDI). Bruininks-Oseretsky Test of Motor Proficiency, 2nd edition (BOT-2).	Patients participated in daily physical therapy, occupational therapy and psychological treatment and received nursing and medical care as necessary.	NRS = *p* < 0.001 FDI = *p* < 0.001 LEFS = *p*< 0.001 COPM = < 0.001 MASC = *p*< 0.001 CDI *p* = 0.003 All BOT-2 domains = *p* < 0.001. Patients underwent any procedures (e.g., nerve blocks) during or immediately prior to participation in the rehabilitation program.	No randomization. No control group. No isolated treatment effects. Uncontrolled prior treatment history in analyses.
Sherry et al., 1999	n = 103	IASP clinical and research criteria.	Visual analog scale (VAS) and Brief Symptom Inventory (BSI) at admission and remission.	An intensive exercise program (most received a daily program of 4 h of aerobic, functionally directed exercises, 1–2 h of hydrotherapy and desensitization). No medications or modalities were used. All had a screening psychological evaluation.	VAS = *p* = 0.021. BSI depression *p* = 0.037. BSI paranoid ideation *p* = 0.048. 1 child (2%) was dysfunctional with CRPS pain, and 5 (10%) had persistent mild pain but were fully functional. Median time between remission of the first episode of CRPS and the start of the second episode = 2 months (range = 2 weeks to 4 years). Predictors of recurrent episodes: previous suicide attempts (*p* = 0.026,), history of an eating disorder (*p* = 0.028).	No long-term follow-up. No control group.
Lee et al., 2002	n = 28. Group A = PT once per week for 6 weeks Group B = PT 3 times per week for 6 weeks.	Wilder et al. criteria.	Pretreatment, at completion of the treatment program and (3) long-term follow-up at 6 to 12 months. Visual analog scale (VAS), Standardized gait impairment score (SGIS), Child Health Questionnaire (CHQ-CF87), Child Depression Inventory (CDI), Revised Children’s Manifest Anxiety Scale (CMAS), Compliance.	Individualized physical therapy. Individualized 6 weekly sessions cognitive behavioral therapy. Standard educational program.	At the short-term follow-up, both groups showed improvement in all five outcome measures related to pain and physical functioning (*p* < 0.001 for all measures with a change in median values). There were no between-group differences in any of these measures at baseline or at either follow-up assessment. 79% compliance good.	Small sample. Not standardized after the 6-week protocol.

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
