# Peer review of "Treatment of Complex Regional Pain Syndrome in Children and Adolescents: A Structured Literature Scoping Review"

_children, 2020, doi:10.3390/children7110245_

Round 1

Reviewer 1 Report

The changed by editing it , it Ok .

Author Response

Thank you for reviewing again the manuscript

Reviewer 2 Report

The authors have done a good job in further revising their manuscript.

One last comment regarding the flowchart: Please correct the number in the flowchart. 240 instead of 24 “records after duplicates removed”. The last part of the flowchart is still not correct. From 103 articles checked for eligibility 81 were excluded resulting in 22 instead of 6 articles. What happened with the remaining 16 articles?

Author Response

Sorry for the uploaded wrong version of the flow-chart. The correct one has been inserted. 

Thank you for you correction

This manuscript is a resubmission of an earlier submission. The following is a list of the peer review reports and author responses from that submission.

Round 1

Reviewer 1 Report

  1. article identification process seems fine but selection criteria seems a bit broad and vague.
  2. key part in comparing outcome of an intervention is if the cohorts/ patient population is similar and the intervention was controlled.
  3. At most the conclusion can be that no therapy can be called effective, as nothing have been evaluated properly.

Author Response

Q1) article identification process seems fine but selection criteria seems a bit broad and vague.

A1) Other selection criteria have been added.

Q2) key part in comparing outcome of an intervention is if the cohorts/ patient population is similar and the intervention was controlled.

A2) Thank you for the comment. The Compatibility of the studies were assessed according to the Newcastle-Ottawa Quality Assessment Scale Cohort Studies and reported in the Appendix.

Q3) At most the conclusion can be that no therapy can be called effective, as nothing have been evaluated properly.

A3) Thank you for the comment. The conclusion was re-written according your suggestions.

Reviewer 2 Report

Thank you for giving me an opportunity of reviewing this paper. It is an interesting paper; however, some concerns can be made.

  1. This paper is not a systematic review, but just a narrative review. There is no result of pooled (combined) data analysis (e.g., forest plot or funnel plot). The authors only describe the result of each study. 
  2. The conclusion is inadequate. The authors concluded that "physical therapy and cognitive-behavioral treatment is the recommended option in the management of pediatric CRPS." However, it is hard to agree with this conclusion without a proper direct-comparison of these treatments with pharmacological treatment.
  3. For a systematic review to have a sound conclusion, the quality of the studies involved must be high. In this paper, the studies included for analysis are retrospective studies or those without placebo-control.
  4. Minor error: page 5, line 3 in the third column --> "scale" seems to be a typo (scale is correct).

Author Response

Q1) This paper is not a systematic review, but just a narrative review. There is no result of pooled (combined) data analysis (e.g., forest plot or funnel plot). The authors only describe the result of each study.

A1) Thank you for your comment. Let me to define a systematic review as a detailed, systematic and transparent means of gathering, appraising and synthesizing evidence to answer a well-defined question (as in the submitted manuscript). Otherwise, a meta-analysis is a statistical procedure for combining numerical data from multiple separate studies (using proper tools as forest plot or funnel plot).

Q2) The conclusion is inadequate. The authors concluded that "physical therapy and cognitive-behavioral treatment is the recommended option in the management of pediatric CRPS." However, it is hard to agree with this conclusion without a proper direct-comparison of these treatments with pharmacological treatment.

Q3) For a systematic review to have a sound conclusion, the quality of the studies involved must be high. In this paper, the studies included for analysis are retrospective studies or those without placebo-control.

A2 and A3) Thank you for your comment. The conclusion was re-written according your suggestions. The treatment direct-comparison discussion was included in the limits of the study. Unfortunately, the literature is poor of high-quality studies about CPRS. In the review were included prospective and randomized trials, the retrospective studies included have a high relevance for the treatment protocol and methodology.

Q4) Minor error: page 5, line 3 in the third column --> "scale" seems to be a typo (scale is correct).

A4) Thank you for your correction

Reviewer 3 Report

This systematic review examines the available literature on the effectiveness of the different treatment options of complex regional pain syndrome (CRPS) in children and adolescents. Such review can be an important contribution to the controversial discussion about the treatment of CRPS and could provide an important scientific evidence for treatment recommendations.

The result of the study is that the combination of physical therapy (PT) and cognitive behavioural therapy (CBT) is the most effective therapy. The added value of the study and the conclusion for practice must be worked out more clearly by the authors. Overall, too little attention is paid to the therapeutic options in detail and to the importance of multimodal therapy.

Altogether, this work – given major revisions - would be an important contribution to the field. Please find more specific comments below.

Abstract

The sentence “The articles focused on physical (PT), cognitive-behavioral (CBT) and 19 pharmacological (PhT) treatments.” is misleading. It needs to be stressed that articles also report on multimodal treatments not on single interventions.

Methods

In the section „methods“ information on a review protocol is missing as well as the registration information including registration number.

A detailed description of the search terms should be provided. The number of identified articles seems very low.

One exclusion criterion was “Diagnosis or differential diagnosis of complex regional pain syndrome type 1”. Could the authors specify the reason for this exclusion criterion? Were studies reporting on treatment but also describing diagnoses excluded?

“Duplicates and articles on other topics, with poor scientific methodology, or without an accessible abstract were not included in the study; abstracts, case reports, conference presentations, editorials and expert opinions were also excluded.”: Please avoid redundant information with defined exclusion criteria, e.g. case reports. How was “poor scientific methodology” identified? How many studies were excluded due to low quality? This information cannot be retrieved from the flowchart.

In section 2.3 it is not clear if two or three researchers extracted data.

“Reference lists from the selected papers were also screened.” This was already described above.

Was the risk of bias assessment performed for all n=103 studies? To the reviewer’s knowledge this only needs to be applied to the final selection of articles. It is not comprehensible which articles are presented in the table in Appendix A.

Results

Flowchart: Why were articles removed before records were screened? Please add the number of excluded articles. Also, double check the articles excluded after full-text screening. Where does the n=7 refer to?

Reporting in Table 1 should be more standardized, e.g., first total N, then gender and age distribution reported in the same style; first time points of data assessment, then measures used. Please revise the table so that the reader can easily compare studies.

Table 1: Why is the CBT treatment in the study by Donado et al, 2017, not reported in the treatment section of the table?

Table 1 should also include information on the CRPS diagnostic criteria used in each study.

The description of studies in the text of the results sections needs a more uniform reporting style; e.g., now, for one study the patient’s age is reported, but not for the other studies.

Discussion

Please start the discussion with a summary of results.

In the discussion the relevance and the heterogeneity of the different outcome measures used in the studies and the sample sizes should be discussed with regard to the different treatment approaches. It makes a difference whether a visual analogue scala (VAS) or a functional disability inventory (FDI) or several different tests are used as an outcome measure. And it has a different meaning whether or not a 4-month follow-up assessment has been carried out.

Currently, the discussion of CRPS diagnostic criteria is very long although it is not recognized in the results section at all.

Limitations of the study should be added.

Conclusions

The conclusion is too general and short. Examples should also be given of what needs to be specifically investigated in further studies. Additionally, the conclusion for practice must be worked out more clearly by the authors.

Author Response

This systematic review examines the available literature on the effectiveness of the different treatment options of complex regional pain syndrome (CRPS) in children and adolescents. Such review can be an important contribution to the controversial discussion about the treatment of CRPS and could provide an important scientific evidence for treatment recommendations.

The result of the study is that the combination of physical therapy (PT) and cognitive behavioural therapy (CBT) is the most effective therapy. The added value of the study and the conclusion for practice must be worked out more clearly by the authors. Overall, too little attention is paid to the therapeutic options in detail and to the importance of multimodal therapy.

Altogether, this work – given major revisions - would be an important contribution to the field. Please find more specific comments below.

Abstract

Q1) The sentence “The articles focused on physical (PT), cognitive-behavioral (CBT) and 19 pharmacological (PhT) treatments.” is misleading. It needs to be stressed that articles also report on multimodal treatments not on single interventions.

A1) Thank you for the comment. The requested changes were made.

Methods

Q2) In the section „methods“ information on a review protocol is missing as well as the registration information including registration number.

A2) Thank you for the comment. Unfortunately, the review was not recorded to any register.

Q3) A detailed description of the search terms should be provided. The number of identified articles seems very low.

One exclusion criterion was “Diagnosis or differential diagnosis of complex regional pain syndrome type 1”. Could the authors specify the reason for this exclusion criterion? Were studies reporting on treatment but also describing diagnoses excluded?

A3) Thank for your comment. The aim of the study was to report the main findings about the treatment of CRPS in children and adolescent. For this reason, in the review were included only studies reporting treatment protocols. No articles describing diagnosis and treatment were found during the screening

Q4) “Duplicates and articles on other topics, with poor scientific methodology, or without an accessible abstract were not included in the study; abstracts, case reports, conference presentations, editorials and expert opinions were also excluded.”: Please avoid redundant information with defined exclusion criteria, e.g. case reports. How was “poor scientific methodology” identified? How many studies were excluded due to low quality? This information cannot be retrieved from the flowchart.

A4) Thank for your comment. The poor scientific methodology papers were included in flowchart as “excluded with reason.”

Q5) In section 2.3 it is not clear if two or three researchers extracted data.

A5) Thank you for the comment. The typo was corrected.

Q6) “Reference lists from the selected papers were also screened.” This was already described above.

A6) Thank you for the comment. The sentence was removed.

Q7) Was the risk of bias assessment performed for all n=103 studies? To the reviewer’s knowledge this only needs to be applied to the final selection of articles. It is not comprehensible which articles are presented in the table in Appendix A.

A7) The risk of bias assessment was performed for 15 articles, after the full-text reading, as reported in Appendix A.

Results

Q8) Flowchart: Why were articles removed before records were screened? Please add the number of excluded articles. Also, double check the articles excluded after full-text screening. Where does the n=7 refer to?

A8) Thank for you comment. Before the first screening the duplicates of articles with the same title and authors were removed, but the studies were included.

The number 7 referred to the articles with poor or fair quality.

Q9) Reporting in Table 1 should be more standardized, e.g., first total N, then gender and age distribution reported in the same style; first time points of data assessment, then measures used. Please revise the table so that the reader can easily compare studies.

The description of studies in the text of the results sections needs a more uniform reporting style; e.g., now, for one study the patient’s age is reported, but not for the other studies.

A9) Thank you for the comment. Table 1 was revised.

Q10) Table 1: Why is the CBT treatment in the study by Donado et al, 2017, not reported in the treatment section of the table?

A10) Thank you for the comment. Donado et al. performed a pharmacological treatment after the CBT approach failure. Unfortunately, the authors did not report the previous CBT protocol.

Q11) Table 1 should also include information on the CRPS diagnostic criteria used in each study.

A11) Thank you for the comment. Diagnostic criteria were added

Discussion

Q12) Please start the discussion with a summary of results.

A12) Thank you for the comment. The key findings were added at the top of the discussion.

Q13) In the discussion the relevance and the heterogeneity of the different outcome measures used in the studies and the sample sizes should be discussed with regard to the different treatment approaches. It makes a difference whether a visual analogue scala (VAS) or a functional disability inventory (FDI) or several different tests are used as an outcome measure. And it has a different meaning whether or not a 4-month follow-up assessment has been carried out.

Limitations of the study should be added.

A13) Thank you for the comment. The requested points of discussion were added in limits of the study.

Q14) Currently, the discussion of CRPS diagnostic criteria is very long although it is not recognized in the results section at all.

A14) Thank you for the comment. The paragraph was summarized.

Conclusions

Q15) The conclusion is too general and short. Examples should also be given of what needs to be specifically investigated in further studies. Additionally, the conclusion for practice must be worked out more clearly by the authors.

A15) Thank you for the comment. The conclusion was amplified and re-written.

Round 2

Reviewer 1 Report

A good effort. Nice review of literature, limitations of current understanding of the pathophysiology of the disease highlighted which is reflective in poor and vague criteria to diagnose this condition and results in poor and vague management.

Reviewer 2 Report

Thank you for the authors' response.
The logic leading to the conclusion is feeble.
As the authors said in their response, a systematic review must have the process of "gathering, appraising and synthesizing evidence." Although the authors gathered six papers (with low quality or small sample size), it is difficult to believe that there has been a logical process of synthesizing evidence. Only a narrative description of the previous studies cannot draw meaningful conclusions with clinical significance.
I do not think this paper can add any new knowledge to the readers.

Reviewer 3 Report

The authors have provided a revised version based on reviewers’ comments. Some comments were not considered.

The manuscript has improved. However, it needs further changes/clarification.

Abstract:

The sentence “The articles focused on physical (PT), associated to cognitive-behavioral (CBT) and pharmacological (PhT) treatments” still is misleading and now implies that PT is the core discipline of treatments. This is not true. The sentence should read: “The articles focused on physical, cognitive-behavioral and pharmacological treatments and a combinations of those different approaches.”

Methods:

The review protocol was not registered which is a core requirement of systematic reviews according to Prisma Guidelines. Because of this and other limitations, e.g. no systematic extraction of treatment effects, the work should not be referred to as a systematic review but instead should be referred to as “scoping review” in the title and throughout the manuscript.

In my previous review I requested that a detailed description of the search terms should be provided. This is crucial because the search strategy is the core piece of a literature review. The authors did not reply to this request. Please add this information as a table or supplementary material for more transparency.

Please change exclusion criterion 4 to “studies solely reporting on diagnosis or differential diagnosis of complex regional pain syndrome type 1.”

Flowchart: the description “excluded with reasons” is not precise. Please rephrase, e.g. excluded due to poor scientific methodology”.

Please standardize reporting in the table, i.e. either number of excluded studies in ( ) or not.

In their response to the initial review authors state that “The risk of bias assessment was performed for 15 articles, after the full-text reading,” To my understanding, 7 articles were excluded due to poor scientific quality. I suppose those were part of the quality appraisal. Why were two additional papers appraised for quality?

Results:

Flowchart: The authors state in their response “Before the first screening the duplicates of articles with the same title and authors were removed, but the studies were included...” As I read the flowchart 264 articles were identified in the two data bases. 24 articles were removed because they were duplicates. After this, only 186 articles were screened. What happened with the remaining 54 articles? Please clarify and specify in the flowchart.

Table 1: instead of 7p. please write n=7

Table 1: measure points are not reported for all studies. Please add this information.

Table 1: In reporting of results the level of detail is very heterogenous. Please standardize the reporting to some extent.

The description of studies in the text of the results sections needs a more uniform reporting style; e.g., now, for one study the patient’s age is reported, but not for the other studies.

Discussion

“According to our data, the physical therapy combined with cognitive behavioral treatment is 142 recommended as the first line approach in the CRPS affected children and adolescent.” Because treatment effects of studies could not be compared, this sentence is overinterpreting results.

 The sentence “despite the major randomized prospective studies were included in the review, no control group article or direct comparisons between treatments have been published.” is not clear. If randomized studies were included, there would be a control group.